# Vid2Game: Controllable Characters Extracted from Real-World Videos

**Oran Gafni**
Facebook AI Research
oran@fb.com

**Lior Wolf**
Facebook AI Research & Tel Aviv Uni.
wolf@fb.com

**Yaniv Taigman**
Facebook AI Research
yaniv@fb.com

## Abstract

We extract a controllable model from a video of a person performing a certain activity. The model generates novel image sequences of that person, according to user-defined control signals, typically marking the displacement of the moving body. The generated video can have an arbitrary background, and effectively capture both the dynamics and appearance of the person.

The method is based on two networks. The first maps a current pose, and a single-instance control signal to the next pose. The second maps the current pose, the new pose, and a given background, to an output frame. Both networks include multiple novelties that enable high-quality performance. This is demonstrated on multiple characters extracted from various videos of dancers and athletes.

## 1 Introduction

We propose a new video generation tool that is able to extract a character from a video, reanimate it, and generate a novel video of the modified scene, see Fig. 1. Unlike previous work, the reanimation is controlled by a low-dimensional signal, such as the one provided by a joystick, and the model has to complete this signal to a high-dimensional full-body signal, in order to generate realistic motion sequences. In addition, our method is general enough to position the extracted character in a new background, which is possibly also dynamic. A video containing a short explanation of our method, samples of output videos, and a comparison to previous work, is provided in `https://youtu.be/sNp6HskavBE`.

Our work provides a general and convenient way for human users to control the dynamic development of a given video. The input is a video, which contains one or more characters. The characters are extracted, and each is associated with a sequence of displacements. In the current implementation, the motion is taken as the trajectory of the center of mass of that character in the frame. This can be readily generalized to separate different motion elements. Given a user-defined trajectory, a realistic video of the character, placed in front of an arbitrary background, is generated.

The method employs two networks, applied in a sequential manner. The first is the Pose2Pose (P2P) network, responsible for manipulating a given pose in an autoregressive manner, based on an input stream of control signals. The second is the Pose2Frame (P2F) network, accountable for generating a high-resolution realistic video frame, given an input pose and a background image.

Each network addresses a computational problem not previously fully met, together paving the way for the generation of video games with realistic graphics. The Pose2Pose network enables guided human-pose generation for a specific trained domain (e.g., a tennis player, a dancer, etc.), where guiding takes the form of 2D motion controls, while the Pose2Frame network allows the incorporation of a photo-realistic generated character into a desired environment.

In order to enable this, the following challenges are to be addressed: (1) replacing the background requires the system to separate the character from the surroundings, which is not handled by previous work, since they either embed the character into the same learned background, or paste the generated character into the background with noticeable artifacts, (2) the separation is not binary, and some effects, such as shadows, blend the character's motion effect with that background information, (3) the control signal is arbitrary, and can lead the character to poses that are not covered by the training set, and (4) generated sequences may easily drift, by accumulating small errors over time.

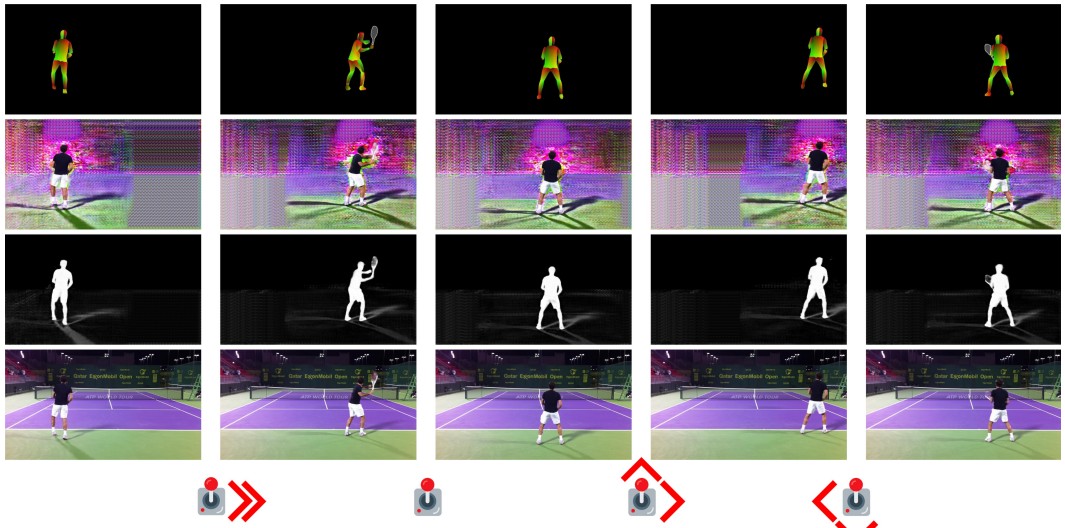

Figure 1: Our method extracts a character from an uncontrolled video, and enables us to control its motion. The pose of the character, shown in the first row, is created by our Pose2Pose network in an autoregressive way, so that the motion matches the control signal illustrated by the joystick. The second row depicts the character's appearance, as generated by the Pose2Frame network, which also generates the masks shown in the third row. The final frame (last row) blends a given background and the generated frames, in accordance with these masks.

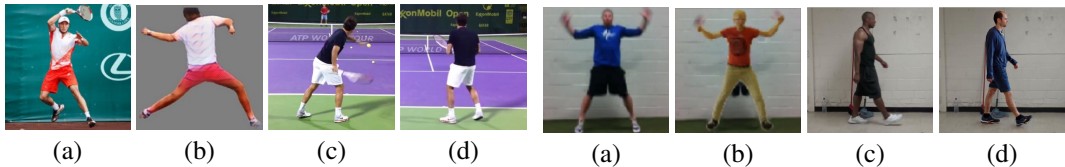

| (a) | (b) | (c) | (d) | (a) | (b) | (c) | (d) |

Figure 2: Comparison with Esser et al. (2018b). (a) Their input, (b) their output, (c) a frame from our training video, (d) our generated frame. With different objectives and dataset types, a direct comparison is not applicable. Qualitatively, Esser et al. (2018b) output a low-res image with noticeable artifacts, and cannot model the racket, while ours is indistinguishable from the source.

Figure 3: Comparison with Esser et al. (2018a). (a) Their input, (b) their generated output, (c) our pose input, (d) the output generated by our P2F network. In contrast to our method, Esser et al. (2018a) do not render environmental effects, resulting in unnatural blending of the character, undesired residues (e.g. source clothing), and works in lower resolution.

Both the Pose2Pose and Pose2Frame networks adopt the pix2pixHD framework of Wang et al. (2018b) as the generator and discriminator backbones, yet add many contributions in order to address the aforementioned challenges. As a building block, we use the pose representation provided by the DensePose framework by Rîza Alp Güler (2018), unmodified. Similarly, the hand-held object is extracted using the semantic segmentation method of Zhou et al. (2019), which incorporates elements from Maninis et al. (2018); Law & Deng (2018).

In addition to the main application of generating a realistic video from a 2D trajectory, the learned Pose2Frame network can be used for other applications. For example, instead of predicting the pose, it can be extracted from an existing video. This allows us to compare the Pose2Frame network directly with recent video-to-video solutions.

## 2  RELATED WORK

Novel view synthesis is a task where unseen frames, camera views, or poses, are synthesized given a prior image. Recent approaches have also shown success in generating detailed images of human subjects in different poses (Balakrishnan et al., 2018; Kanazawa et al., 2018), where some of them

also condition on pose (Chan et al., 2018; Yang et al., 2018; Li et al., 2019) to guide the generation. These approaches do not build a movable character model, but transfers one image to target poses. The pose variability in these images is smaller than required for our application, the handling of the background is limited, and these were also not demonstrated on video. For example, much of the literature presents results on a fashion dataset, in which the poses are limited and a white background is used. Another common benchmark is gathered from surveillance cameras, where the resolution is low, and background generation is lacking due to an inherent lack of supervision.

A method for learning motion patterns by analyzing YouTube videos is demonstrated by Peng et al. (2018), where synthetic virtual characters are set to perform complex skills in physically simulated environments, leveraging a data-driven Reinforcement Learning method that utilizes a reference motion. This method outputs a control policy that enables the character to reproduce a particular skill observed in video, which the rendered character then imitates. Unlike our method, the control signal is not provided online, one frame at a time. In addition, rendering is performed using simulated characters only, and the character in the video is not reanimated.

Autoregressive models, which can be controlled one step at a time, are suitable for the dynamic nature of video games. However, such models, including RNNs, can easily drift with long range sequences (Fragkiadaki et al., 2015), and training RNN models for long sequences suffers from vanishing or exploding gradients. Holden et al. (2017) propose a more stable model by generating the weights of a regression network at each frame as a function of the motion phase. However, this is mostly practical to apply given a limited number of keypoints, whereas dense pose models contain more information.

Generative Adversarial Networks (GANs) (Goodfellow et al., 2014) and conditional GANs (Mirza & Osindero, 2014), have been used for video synthesis by Vondrick et al. (2016) who separately generates the static background and the foreground motion. Frameworks such as vid2vid (Wang et al., 2018a; Chan et al., 2018) learn mappings between different videos, and demonstrate motion transfer between faces, and from poses to body. In these contributions, the reference pose is extracted from a real frame, and the methods are not challenged with generated poses. Working with generated poses, with the accompanying artifacts and the accumulated error, is considerably more challenging. In order to address this, we incorporate a few modifications, such as relying on a second input pose, in case one of the input poses is of lesser quality, and add additional loss terms to increase the realism of the generated image. In addition, these approaches model the entire frame, including both the character and the background, which usually leads to blurry results (Pumarola et al., 2018; Chao et al., 2018), particularly near the edges of the generated pose, and with complex objects, such as faces. It also leads to a loss of details from the background, and to unnatural background motion.

A method for mixing the appearance of a figure seen in an image with an arbitrary pose is presented by Esser et al. (2018b). While it differs greatly in the performed task, we can compare the richness of the generated images, as shown in Fig. 2. Their method results in a low-resolution output with noticeable artifacts, and cannot model the object, while our result is indistinguishable from the source. The same is true for the follow-up work (Esser et al., 2018a). We work at a higher resolution of 1024p, while their work is limited to low-resolution characters, see Fig. 3. Similarly, the work of Balakrishnan et al. (2018) provides lower resolution outputs, limited to the same background, and does not handle shadows (as seen in Fig. 9-10 of that work).

In another set of experiments, Esser et al. (2018a) also present a step toward our task and show results for generating a controllable figure, building upon the phase-based neural network of Holden et al. (2017). Their work is keypoint based and does not model environmental factors, such as shadows. The videos presented by Esser et al. (2018a) for a controllable figure are displayed only on a synthetic background with a checkerboard floor pattern in an otherwise empty scene. These examples are limited to either walking or running, and the motion patterns are of an existing animation model.

## 3 METHOD OVERVIEW

The method's objective is to learn the character's motion from a video sequence, such that new videos of that character can be rendered, based on a user-provided motion sequence. The input of the training procedure is a video sequence of a character performing an action. From this video, the pose and an approximated foreground mask are extracted by the DensePose network, augmented by the semantic

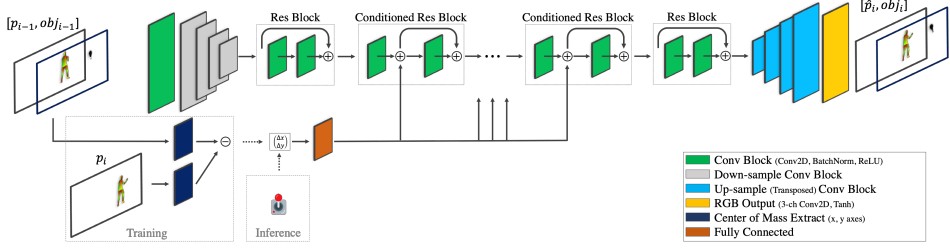

Figure 4: The architecture of the Pose2Pose generator. During training, the middle $n_r - 2$ residual blocks are conditioned by a linear projection (FC layer) of the center-mass differences between consecutive frames (in the x and y axes). For each concatenation of input pose and object $[p_{i-1}, obj_{i-1}]$, the network generates the next consecutive pose and object $[p_i, obj_i]$. At inference time, the network generates the next pose-object pair in an autoregressive manner, conditioned on input directions.

segmentation of the hand-held object, for each frame. The trajectory of the center of mass is taken to be the control sequence. At test time, the user provides a sequence of 2D displacements, and a video is created, in which the character moves in accordance with this control sequence. The background can be arbitrary, and is also selected by the user. The method then predicts the sequence of poses based on the given control sequence (starting with an arbitrary pose), and synthesizes a video in which the character extracted from the training video is rendered in the given background.

The following notation is used: a video sequence with frames $f_i$ is generated, based on a sequence of poses $p_i$ and a sequence of background images $b_i$, where $i = 1, 2, \ldots$ is the frame index. The frame generation process also involves a sequence of spatial masks $m_i$ that determine which regions of the background are replaced by synthesized image information $z_i$.

To generate a video, the user provides the pose at time zero: $p_0$, the sequence of background images $b_i$ (which can be static, i.e., $\forall i \; b_i = b$) and a sequence of control signals $s_i$. In our experiments, the control signal is typically comprised of the desired 2D displacement of the animated character.

Our method is an autoregressive pose model, coupled with a frame-rendering mechanism. The first aspect of our method creates a sequence of poses, and optionally of hand-held objects. Each pose and object pair $[p_i, obj_i]$ is dependent on the previous pair $[p_{i-1}, obj_{i-1}]$, as well as on the current control signal $s_i$. The second aspect generates the current frame $f_i$, based on the current background image $b_i$, the previous combined pose and object $p_{i-1} + obj_{i-1}$, and the current combined pose and object $p_i + obj_i$. The pose and object are combined by simply summing the object channel with each of the three RGB channels that encode the pose. This rendering process includes the generation of both a raw image output $z_i$ and a blending mask $m_i$. $m_i$ has values between 0 and 1, with $1 - m_i$ denoting the inverted mask.

Formally, the high-level processing is given by the following three equations:

$$[p_i, obj_i] = P2P([p_{i-1}, obj_{i-1}], s_i) \tag{1}$$

$$(z_i, m_i) = P2F([p_{i-1} + obj_{i-1}, p_i + obj_i]) \tag{2}$$

$$f_i = z_i \odot m_i + b_i \odot (1 - m_i) \tag{3}$$

where $P2P$ and $P2F$ are the Pose2Pose and the Pose2Frame networks. As stated, $P2F$ returns a pair of outputs that are then linearly blended with the desired background, using the per-pixel multiplication operator $\odot$.

## 4 THE POSE2POSE NETWORK

As mentioned, the P2P network is an evolution of the pix2pixHD architecture. Although the primary use of the pix2pixHD framework in the literature is for unconditioned image-to-image translation, we show how to modify it to enable conditioning on a control signal. The P2P network generates a scaled-down frame (512 pixels wide), allowing the network to focus on pose representation, rather than high-resolution image generation. Generation of a high-res output is deferred to the P2F network. This enables us to train the P2P network much more effectively, resulting in a stable training process that generates natural dynamics, and leads to significantly reduced inference time (post-training).

The generator's architecture is illustrated in Fig. 4. The encoder is composed of a convolutional layer, followed by convolutions with batch normalization Ioffe & Szegedy (2015) and ReLU Nair & Hinton (2010) activations. The latent space combines a sequence of $n_r$ residual blocks. The decoder is composed of fractional strided convolutions with instance normalization Ulyanov et al. (2016) and ReLU activations, followed by a single convolution terminated by a Tanh activation for the generated frame output.

Recall that the P2P network also receives the control signal as a second input (Eq. 1). In our experiment, the control signal is a vector of dimension $n_d = 2$ representing displacements along the $x$ and $y$ axes. This signal is incorporated into the network, by conditioning the center $n_r - 2$ blocks of the latent space.

The conditioning takes place by adding to the activations of each residual block, a similarly sized tensor that is obtained by linearly projecting the 2D control vector $s_i$.

**Modified conditional block** Rather than applying a conditioning block based on a traditional ResNet block, we apply a modified one that does not allow for a complete bypass of the convolutional layers. This form of conditioning increases the motion naturalness, as seen in our ablation study.

The specific details are as follows. The P2P network contains a down-sampling encoder $e$, a latent space transformation network $r$, and an up-sampling decoder $u$. The $r$ network is conditioned on the control signal $s$, and contains $n_r$ blocks of two types: vanilla residual blocks ($v$), and conditioned blocks $w$.

$$P2P(p, s) = u(r(e(p), s)) \quad (4) \qquad r = v \circ \underbrace{w \circ w \cdots \circ w}_{n_r - 2 \text{ times}} \circ v \quad (5)$$

The architecture and implementation details of the P2P network can be found in appendix A. Briefly, let $x$ denote the activations of the previous layer, and $f_1(x)$, $f_2(x)$ be two consecutive convolutional layers. Let $s$ be a 2D displacement vector, and $g$ a fully-connected layer with a number of output neurons that equals the product of the dimensions of the tensor $x$. The two block types take the form:

$$v(x) = f_2(f_1(x)) + x \quad (6) \qquad w(x, s) = f_2(f_1(x) + g(s)) + f_1(x) + g(s) \quad (7)$$

## 4.1 TRAINING THE POSE PREDICTION NETWORK

Following Wang et al. (2018b), we employ two discriminators (low-res and high-res), indexed by $k = 1, 2$. During training, the LSGAN (Mao et al., 2017) loss is applied to the generator and discriminator. An L1 feature-matching loss is applied over the discriminators' activations, and a trained VGG (Simonyan & Zisserman, 2014b) network. The loss applied to the generator can then be formulated as:

$$\mathcal{L}_{P2P} = \sum_{k=1}^{2} \left( \mathcal{L}_{LS^k} + \lambda_D \mathcal{L}_{FM_D^k} \right) + \lambda_{VGG} \mathcal{L}_{FM_{VGG}} \quad (8)$$

where the networks are trained with $\lambda_D = \lambda_{VGG} = 10$. The LSGAN generator loss is (the $obj_i$ elements are omitted for brevity):

$$\mathcal{L}_{LS^k} = \mathbb{E}_{(p_{i-1}, s_i)} \left[ (D_k(p_{i-1}, P2P(p_{i-1}, s_i)) - \mathbb{1})^2 \right] \quad (9)$$

The expectation is computed per mini-batch, over the input pose $p_{i-1}$ and the associated $s_i$. The discriminator-feature matching-loss compares the predicted pose with that of the generated pose, using the activations of the discriminator, and is calculated as:

$$\mathcal{L}_{FM_D^k} = \mathbb{E}_{(p_{i-1}, p_i)} \sum_{j=1}^{M} \frac{1}{N_j} ||D_k^{(j)}(p_{i-1}, p_i) - D_k^{(j)}(p_{i-1}, P2P(p_{i-1}, s_i))||_1 \quad (10)$$

with $M$ being the number of layers, $N_j$ the number of elements in each layer, $p_{i-1}$ the input (previous) pose, $p_i$ the current (real) pose, $P2P(p_{i-1}, s)$ the estimated pose, and $D_k^{(j)}$ the activations of discriminator $k$ in layer $j$.

The VGG feature-matching loss is calculated similarly, acting as a perceptual loss over a trained VGG classifier:

$$\mathcal{L}_{FM_{VGG}} = \sum_{j=1}^{M} \frac{1}{N_j'} ||VGG^{(j)}(p_i) - VGG^{(j)}(P2P(p_{i-1}, s_i))||_1 \quad (11)$$

with $N'_j$ being the number of elements in the $j$-th layer, and $VGG^{(j)}$ the VGG classifier activations at the $j$-th layer. The loss applied to the discriminator is formulated as:

$$\mathcal{L}_{D^k} = \frac{1}{2}\mathbb{E}_{(p_{i-1},s_i)}\left[(D_k(p_{i-1}, P2P(p_{i-1}, s_i)))^2\right] + \frac{1}{2}\mathbb{E}_{(p_{i-1},p_i)}\left[(D_k(p_{i-1}, p_i) - \mathbb{1})^2\right] \quad (12)$$

The training sequences are first processed by employing the DensePose network, in order to extract the pose information from each frame. This pose information takes the form of an RGB image, where the 2D RGB intensity levels are a projection of the 3D (I)UV mapping.

By applying a binary threshold over the DensePose RGB image, we are able to create a binary mask for the character in the video. From the binary mask $t_i$ of each frame $i$, we compute the center of mass of the character $\rho_i$. The control signal during training is denoted as $s_i = \rho_i - \rho_{i-1}$.

Due to the temporal smoothness in the videos, the difference between consecutive frames in the full frame-rate videos (30fps) is too small to observe significant motion. This results in learned networks that are biased towards motionless poses. Hence, we train with $\Delta = 2$ inter-frame intervals (where $\Delta = 1$ describes using consecutive frames). During inference, we sample at 30fps and apply a directional conditioning signal that has half of the average motion magnitude during training.

**Stopping criteria** We use the Adam optimizer (Kingma & Ba, 2016) with a learning rate of $2 \cdot 10^{-4}$, $\beta_1 = 0.5$ and $\beta_2 = 0.999$. We observe that training the P2P network does not provide for monotonic improvement in output quality. We stipulate the P2P network final model to be that which yields the lowest loss, in terms of discriminator feature-matching. While there are several losses applied while training the P2P network, the discriminator feature-matching loss is the only one that holds both motion context (i.e. receives both the previous and current pose), and information of different abstraction levels (i.e. feature-matching is applied over different levels of activations). This results in improved motion naturalness, and reduced perceptual distance, as evident from the ablation study.

**Random occlusions** To cope with pose detection imperfections that occasionally occur, which in turn impair the quality of the generated character, we employ a dedicated data augmentation method, in order to boost the robustness of the P2P network. A black ellipse of random size and location is added to each input pose frame within the detection bounding box, resulting in an impaired pose (see appendix Fig. 8), with characteristics that are similar to "naturally" occurring imperfections.

## 5 THE POSE2FRAME NETWORK

While the original pix2pixHD network transforms an entire image to an output image of the same size from a specified domain, our Pose2Frame network transforms a pose to a character that is localized in a specific part of the output image and embedded in a given, possibly dynamic, background. This is done by both refocusing the discriminators' receptive field, and applying a learned blending mask over the raw image output. The DensePose network plays a crucial role, as it provides both the relevant image region and a prior over the blending mask.

Focusing the discriminator on the character eliminates the need for feature-enhancing techniques, such as the introduction of a face-GAN, as done by Chan et al. (2018)), or adding a temporal loss (which is useful for reducing irrelevant background motion) as done by Wang et al. (2018a).

The generator architecture is illustrated in Fig. 5(a). The P2F low-level network architecture details are somewhat similar to those of the P2P network, with the following modifications: (1) the P2F network generates frames with a resolution width of 1024, (2) no conditioning is applied, i.e., the $w$ blocks are replaced by $v$ blocks, (3) the network generates two outputs: the raw image data $z$ and a separate blending mask $m$, (4) the discriminators are altered to reflect the added focus, and (5) new regularization terms are added to ensure that the masking takes place at the relevant regions (Eq. 17), see Fig. 9 in the appendix.

The generated mask $m$ blends the raw output $z$ with the desired background $b$, rendering the final output frame $f$, according to Eq. 3 (omitting the index $i$ for brevity). Note that the blending mask is not binary, since various effects such as shadows, contain both character-derived information and background information, see Fig. 6. Nevertheless, we softly encourage the blending mask to favor the background in regions external to the character, and discourage the generator from rendering meaningful representations outside the character. This is done by employing several regularization

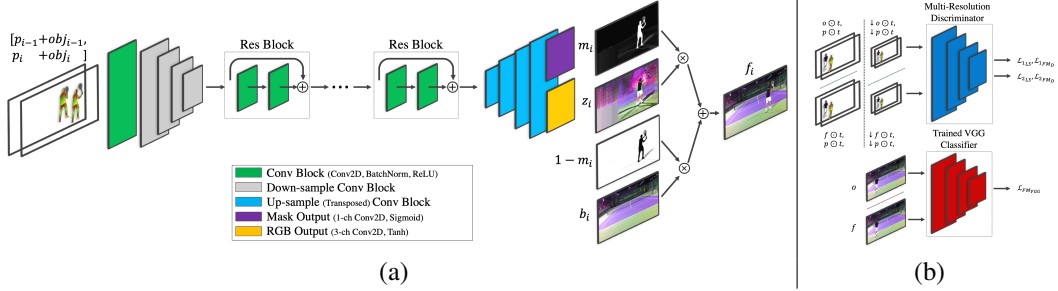

(a)                                                    (b)

Figure 5: The Pose2Frame network. (a) For each two combined input pose and object ($p = [p_{i-1} + obj_{i-1}, p_i + obj_i]$), the network generates an RGB image ($z_i$) and a mask ($m_i$). The RGB and background images are then linearly blended by the generated mask to create the output frame $f_i$. (b) The P2F discriminator setup. The multi-scale discriminator focuses on the binary-thresholded character, obtained with the binary mask $t$, as it appears in both the ground truth image $o$ and the output of the P2F network, for a given pose $p = (p_i, p_{i-1})$. The $\downarrow$ operator denotes downscaling by a factor of two, obtained by average pooling, as applied before the low-resolution discriminator. The VGG feature-matching loss term engages with the full frame, covering perceptual context in higher abstraction layers (e.g. generated shadows).

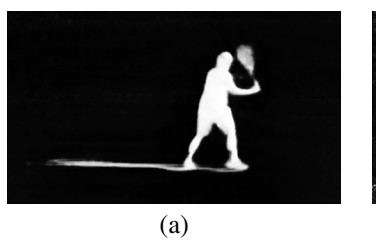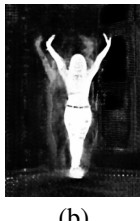

(a)                                (b)

Figure 6: Samples of masks that model both the character, and places in the scene in which appearance is changed by the character. (a) The shadow and the tennis racket of the character are captured by the mask, (b) the dancer's shadow appears as part of the mask.

terms over the generated mask. As a side effect of these added losses, the network is required to perform higher-level reasoning and not rely on memorization. In other words, instead of expanding the mask to include all background changes, the network separates between character dependent changes, such as shadows, held items, and reflections, and those that are independent.

The discriminator setup is illustrated in Fig. 5(b). The discriminator's attention is predominantly shifted towards the character, by applying an inverse binary mask over the character. The masked character image is fed into the discriminators, affecting both the multi-scale loss, and the feature-matching loss applied over the discriminators' activations. In parallel, the fully generated frame is fed into the VGG network, allowing the VGG feature-matching loss to aid in the generation of desired structures external to the character.

### 5.1 TRAINING THE POSE TO FRAME NETWORK

The P2F generator loss is formulated as:

$$\mathcal{L}_{P2F} = \sum_{k=1}^{2} \left( \mathcal{L}_{LS^k} + \lambda_D \mathcal{L}_{FM_D^k} \right) + \lambda_1 \mathcal{L}_{FM_{VGG}} + \lambda_2 \mathcal{L}_{mask} \tag{13}$$

where $\lambda_1 = 10$ and $\lambda_2 = 1$. The LSGAN generator loss is calculated as:

$$\mathcal{L}_{LS^k} = \mathbb{E}_{(p,t)} \left[ \left( D_k(p \odot t, f \odot t) - 1 \right)^2 \right] \tag{14}$$

where $p = [p_{i-1} + obj_{i-1}, p_i + obj_i]$ denotes the two pose images, and $t$ is the binary mask obtained by thresholding the DensePose image at time $i$. The discriminator-feature matching-loss is calculated as:

$$\mathcal{L}_{FM_D^k} = \mathbb{E}_{(p,o,t)} \sum_{j=1}^{M} \frac{1}{N_j} ||D_k^{(j)}(p \odot t, o \odot t) - D_k^{(j)}(p \odot t, f \odot t)||_1, \tag{15}$$

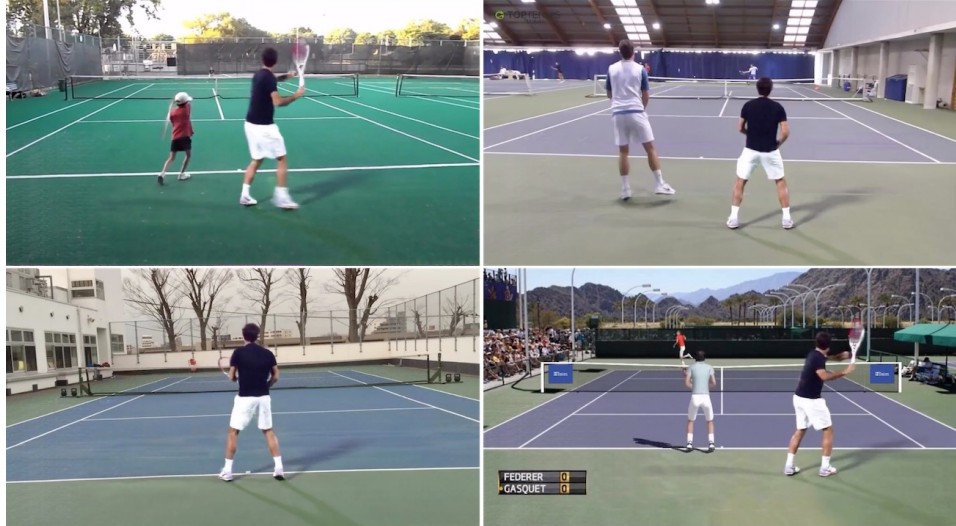

Figure 7: Generated frames for the controllable tennis character, blended into different backgrounds.

with $M$ being the number of layers, $N_j$ the number of elements in each layer, and $o$ the real (ground truth) frame. The VGG feature-matching loss is calculated over the full ground truth frame, rather than the one masked by $t$:

$$\mathcal{L}_{FM_{VGG}} = \sum_{j=1}^{M} \frac{1}{N_j} ||VGG^{(j)}(o) - VGG^{(j)}(f)||_1 \qquad (16)$$

with $o$ being the ground truth frame, $N_j$ being the number of elements in the $j$-th layer, and, as before, $VGG^{(j)}$ the VGG activations of the $j$-th layer.

The mask term penalizes the mask (see appendix Fig. 9 for a visual illustration):

$$\mathcal{L}_{mask} = ||m \odot (1-t)||_1 + ||m_x \odot (1-t)||_1 + ||m_y \odot (1-t)||_1 + ||1 - m \odot t||_1 \qquad (17)$$

where $m$ is the generated mask, and $m_x$ and $m_y$ the mask derivatives in the x and y axes respectively. The first term acts to reduce the mask's activity outside the regions detected by DensePose. The mask, however, is still required to function in such regions, e.g., to render shadows. Similarly, we suppress the mask derivative outside the pose-detected region, in order to eliminate secluded points, and other high-frequency patterns. Finally, a term is added to encourage the mask to be active in the image regions occupied by the character.

The loss applied to the two discriminators is given by:

$$\mathcal{L}_{D^k} = \frac{1}{2}\mathbb{E}_{(p,t)}\left[(D_k(p \odot t, f \odot t))^2\right] + \frac{1}{2}\mathbb{E}_{(p,o,t)}\left[(D_k(p \odot t, o \odot t) - 1)^2\right] \qquad (18)$$

The Adam optimizer is used for P2F similar to the P2P. The training progression across the epochs is visualized in the appendix (Fig. 10).

## 6 EXPERIMENTS

The method was tested on multiple video sequences, see the supplementary video (`https://youtu.be/sNp6HskavBE`). The first video shows a tennis player outdoors, the second video, a person swiping a sword indoors, and the third, a person walking. The part of the videos used for training consists of 5.5min, 3.5min, and 7.5min, respectively. In addition, for comparative purposes, we trained the P2F network on a three min video of a dancer, which was part of the evaluation done by Wang et al. (2018a).

The controllable output of the tennis player is shown in Fig. 1, which depicts the controller signal used to drive the pose, as well as the generated pose $p_i$, object $obj_i$, mask $m_i$, raw frame $z_i$, and

| Dataset | Method | SSIM | LPIPS (SqzNet) | LPIPS (AlexNet) | LPIPS (VGG) |
|---|---|---|---|---|---|
| Tennis | ours | **240**±2 | **265**±3 | **400**±4 | **474**±5 |
| | pix2pixHD | 301±26 | 379±35 | 533±42 | 589±32 |
| Walking | ours | **193**±133 | **216**±149 | **365**±252 | **374**±258 |
| | pix2pixHD | 224±156 | 308±224 | 485±347 | 434±303 |
| Fencing | Ours | **45**±4 | **41**±8 | **52**±11 | **150**±15 |
| | pix2pixHD | 308±95 | 531±129 | 670±168 | 642±86 |

Table 1: Comparison with pix2pixHD (see also Fig. 14). SSIM and LPIPS (multiplied by 1000) are shown for three scenarios: (1) tennis (contains dynamic elements, e.g. other players, crowd, difference in camera angle), (2) walking (different character clothing, lighting, and camera angle), (3) fencing (fixed background and view).

output frame $f_i$. A realistic character is generated with some artifacts (see supplementary video) around the tennis racket, for which the segmentation of the training video is only partially successful. Fig. 7 depicts additional results, in which the character is placed on a diverse set of backgrounds containing considerable motion. Appendix B also present a controlled walking character, and a controlled fencing character, which also appear in the supplementary video.

A comparison of the P2F network with the pix2pixHD method of Wang et al. (2018b) is provided in Tab. 1, and as a figure in appendix Fig. 14. We compare by Structural Similarity (SSIM) (Wang et al., 2004) and Learned Perceptual Image Patch Similarity (LPIPS) Zhang et al. (2018) distance methods. The mean and standard deviation are calculated for each generated video. The LPIPS method provides a perceptual distance metric, by comparing the activations of three different network architectures, VGG (Simonyan & Zisserman, 2014a), AlexNet (Krizhevsky, 2014), and SqueezeNet (Iandola et al., 2016), with an additional linear layer set on top of each network. For each dataset, we select a test set that was not used during training. Although this test set is evaluated as the ground-truth, there is a domain shift between the training and the test video: the tennis test set contains dynamic elements, such as other players, crowd, and a slight difference in camera angle; the walking test set contains different character clothing, background lighting, and camera angle. The fencing test set is more similar to the training set. As seen in appendix Fig. 14, the baseline method results in many background and character artifacts, and a degradation in image and character quality, as it is forced to model the entire scene, rather than focus solely on the character and its shadow, as our method does. This is also apparent in the statistics reported in the table.

Another experiment dedicated to the P2F network (other methods do not employ P2P), compares it with the vid2vid method of Wang et al. (2018a). The results are reported in the supplementary video, and in appendix C. Our method produces far fewer background distortions, can better handle variation in the character's location, and has the ability to embed the character into novel backgrounds.

An ablation study is presented in appendix D, showing the contribution of the various components of the system both quantitatively and qualitatively. In addition, we describe the unfavorable results obtained when replacing the autoregressive model with a concatenative model.

## 7 CONCLUSIONS

Generating smooth motion that combines unpredictable control, the current pose, and previous motion patterns is a challenging task. The proposed novel method employs two autoencoders: one generates autoregressive motion for a specific learned style, and the other generates a realistic frame for blending with a dynamic background.

Our work paves the way for new types of realistic and personalized games, which can be casually created from everyday videos. In addition, controllable characters extracted from YouTube-like videos can find their place in the virtual worlds and augmented realities. The work is still limited in various aspects, such as not allowing control over the illumination of the character, the lack of support for novel views, and not modeling the character's interaction with scene objects.

## ACKNOWLEDGMENTS

The authors would like to thank Lisa Rhee, Ilkka Hartikainen, and Adrian Bryant for allowing us to use their videos for training.

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

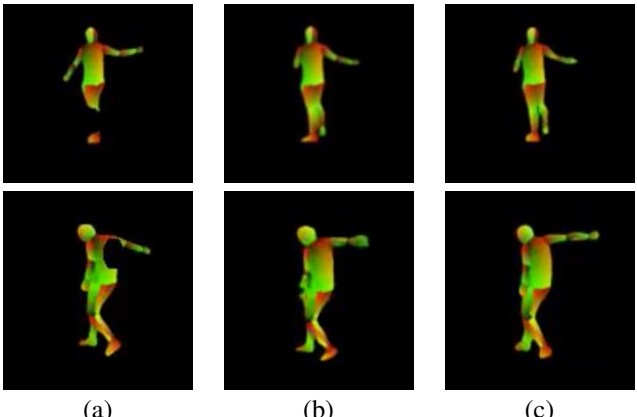

(a)     (b)     (c)

Figure 8: The occlusion-based augmentation technique used to increase robustness during training the P2P network. Each row is a single sample. (a) $p_{i-1}$ with part of it occluded by a random ellipse, (b) the predicted pose $\hat{p}_i$, (c) the ground truth pose $p_i$. The generated output seems to "fill in" the missing limbs, as well as predict the next frame. In this figure and elsewhere, the colors represent the 3D UV mapping.

## A    ADDITIONAL POSE2POSE NETWORK ARCHITECTURE AND IMPLEMENTATION DETAILS

We follow the naming convention of (Wang et al., 2018b; Zhu et al., 2017; Johnson et al., 2016). Let Ck denote a Conv-InstanceNorm-ReLU layer with k filters, each with a kernel size of 7x7, with a stride of 1. Dk denotes a Convolution-InstanceNorm-ReLU layer with $k$ filters and a stride of 2, where reflection padding is used. Vk denotes a vanilla residual block with two 3x3 convolutional layers with the same number of filters on both layers. Wk denotes a conditioned residual block. Uk denotes a 3x3 Fractional-Strided-Convolution-InstanceNorm layer with $k$ filters, and a stride of $0.5$.

The generator, i.e., the P2P network, can then be described as: C64, D128, D256, D512, D1024, V1024, W1024, W1024, W1024, W1024, W1024, W1024, W1024, V1024, U512, U256, U128, U64, C3.

The input images are scaled to a width size of 512 pixels, with the height scaled accordingly.

The discriminators are two PatchGANs (Isola et al., 2017) with an identical architecture of C64,C128,C256,C512, working at the input resolution and a lower resolution, down-sampled by an average-2D-pooling operation with a kernel size of 3, and a stride of 2.

The architecture of the P2F network is similar to that of the P2P network, with the following adjustments: (i) the conditional residual blocks are replaced by non residual ones, (ii) the input of P2F has 6 channels for $p_i$ and $p_{i-1}$, (iii) there is an additional head generating the mask output, which uses a sigmoid activation function.

## B    ADDITIONAL IMAGES

Fig. 8 depicts the random occlusion process (P2P training), in which a black ellipse of random size and location is added to each input pose frame within the detection bounding box. This results in an impaired pose, with characteristics that are similar to "naturally" occurring imperfections.

The mask loss term $\mathcal{L}_{mask}$ of P2F (Sec. 5) is illustrated in Fig. 9.

Fig. 10 depicts the progression during training of the P2F dancer model. As training progresses, the details of the dancer become sharper and the hair becomes part of the mask, despite being outside the DensePose detected area (i.e., off pixels in $t$).

Fig. 11 depicts a controlled walking character along with the control signal and the generated poses.

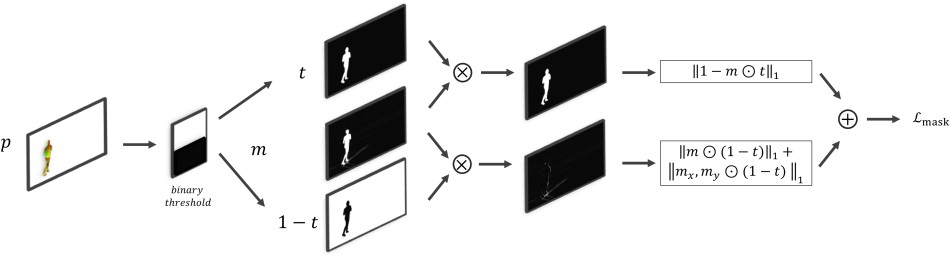

Figure 9: Mask losses applied during the P2F network training. An inverse binary-thresholded mask is used to penalize pixel intensity for the generated mask, in the regions excluding the character of interest. For the generated mask, we apply regularization over the derivatives in the x and y axes as well, to encourage smooth mask generation, and discourage high-frequency pattern generation.

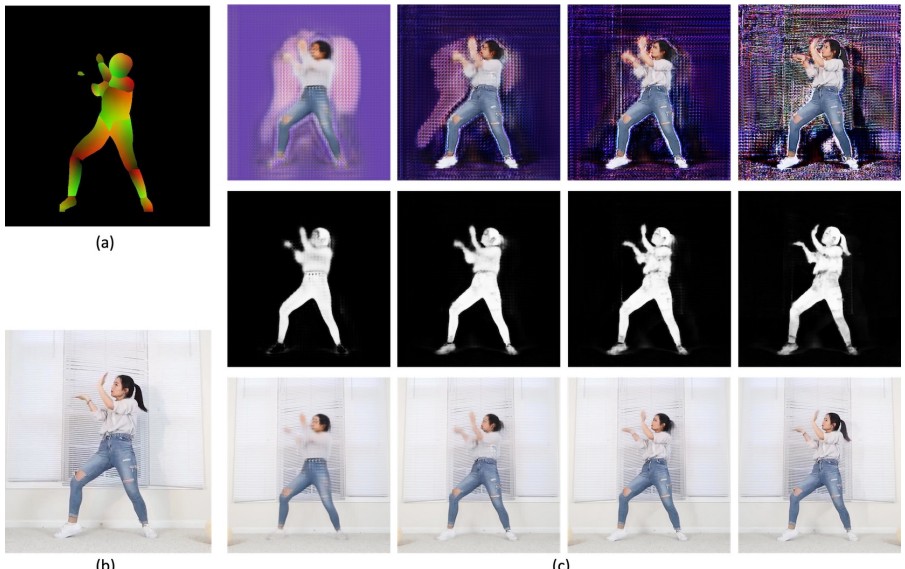

Figure 10: Training the P2F network. (a) A sample pose, (b) the target frame, (c) the generated raw frame, the mask, and the output frame at different epochs: 10, 30, 50, and 200 (final).

The fencing character is shown in Fig. 12. The mask for various frames in the controlled sequence is shown, as well as two backgrounds: the background of the reference video, and an animated background. Fig. 13 depicts an additional controlled walking character, along with the control signal and the generated poses.

Fig 14 compares visually with the baseline method of pix2pixHD (Wang et al., 2018b). As can be seen, the baseline method results in many background and character artifacts, a degradation in image and character quality, as it is forced to model the entire scene, rather than focus solely on the character and the environmental factors, such as in our method.

## C    COMPARISON WITH VID2VID

Fig. 15(a-e) presents a comparison with the method of Wang et al. (2018a). Shown are the target image from which the driving pose is extracted, the extracted pose, the results of the baseline method, and our result. As can be seen, our method handles the background in a way that creates far fewer distortions, as we apply a learned mask, thus background generation is not required.

The characters themselves are mostly comparable in quality, despite our choice not to add a dedicated treatment to the face. In addition, despite not applying considerable emphasis on temporal consistency

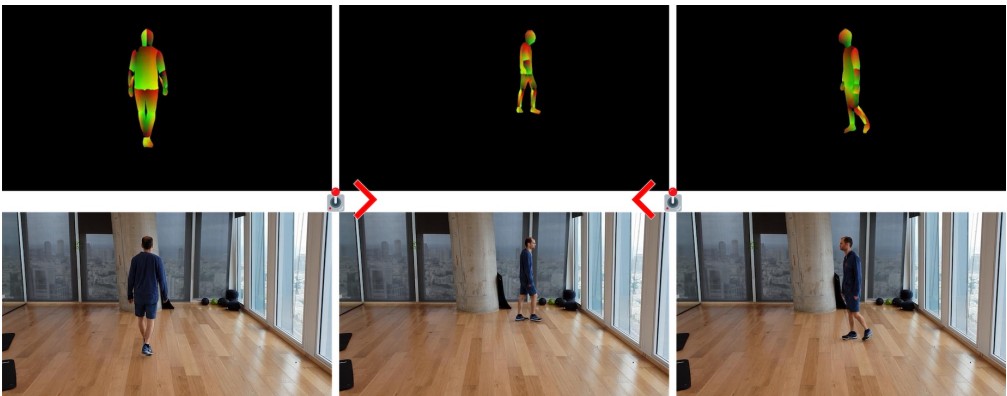

Figure 11: Synthesizing a walking character, emphasizing the control between the frames. Shown are the sequence of poses generated in an autoregressive manner, as well as the generated frames.

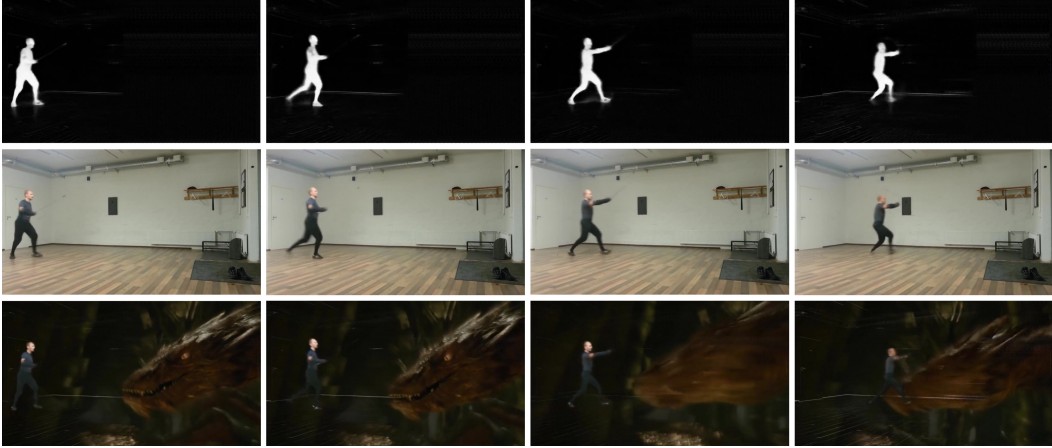

Figure 12: Generated frames for the controllable fencing character. Each column is a different pose. The rows are the obtained mask, and the placement on two different backgrounds: the one obtained by applying a median filter to the reference video, and one taken from a motion picture.

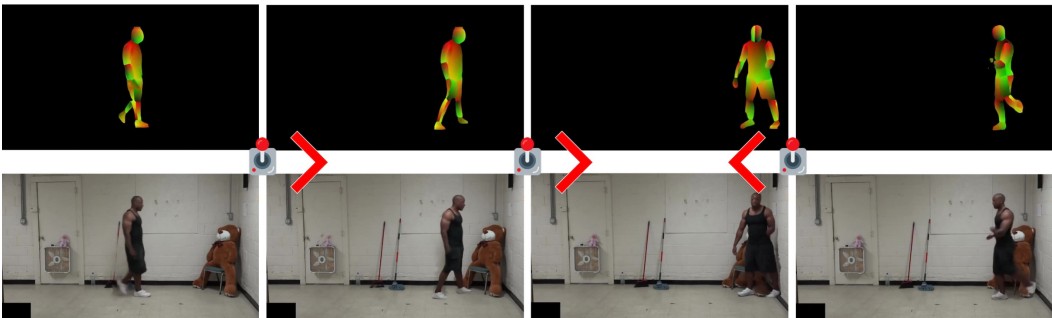

Figure 13: Synthesizing an additional walking character. Shown are the sequence of poses generated in an autoregressive manner, as well as the generated frames.

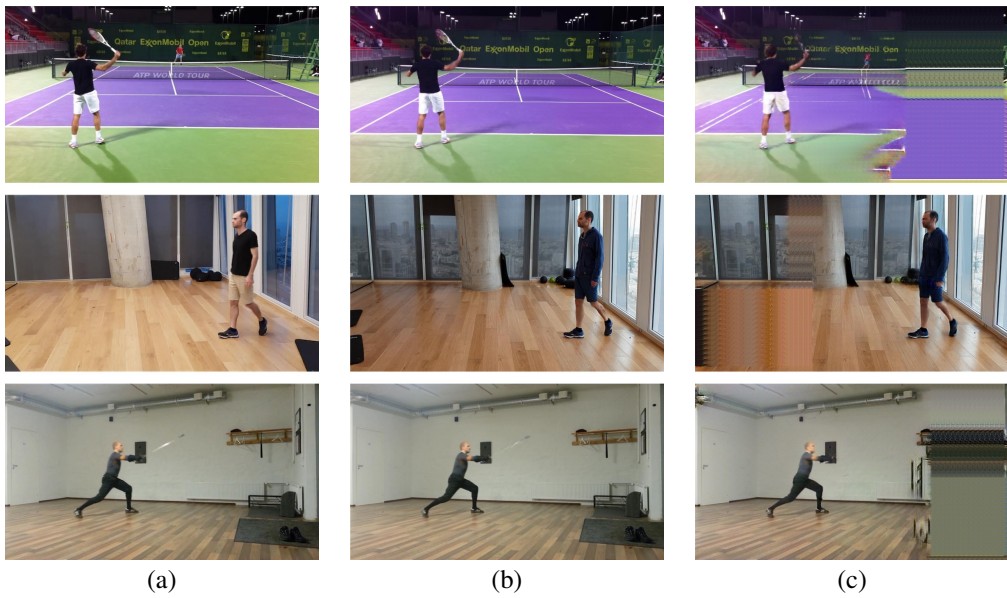

Figure 14: A comparison of the P2F network with the pix2pixHD method of Wang et al. (2018b). (a) Ground truth image used as the pose source, (b) our result, (c) The results of pix2pixHD. The baseline method results in many background artifacts, as it generates the entire frame. The degradation in image quality is apparent as well, and that of the character in particular.

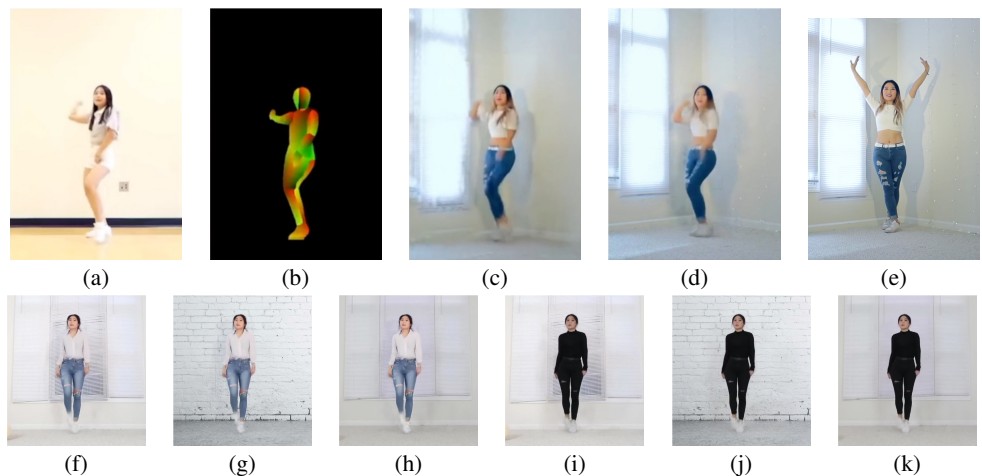

Figure 15: A comparison of the P2F network with the vid2vid method of Wang et al. (2018a). (a) The target-pose image, (b) the pose extracted from this image, (c) the result of vid2vid, (d) our result, (e) a frame from the reference video. Many artifacts are apparent in the background produced by vid2vid. vid2vid also distorts the character's appearance and dimensions to better match the pose. (f-k) The same pose, displayed by two characters on three different backgrounds, demonstrates our advantage over vid2vid in replacing backgrounds.

during training (e.g. optical flow, temporal discriminator), our method produces videos that are as smooth. Finally, the proportions of the character in our video are better maintained, while in the baseline model, the character is slightly distorted toward the driving pose.

In addition, as we demonstrate in Fig. 15(f-k), our method has the ability to replace the background.

| Network Component | SSIM | LPIPS (SqzNet) | LPIPS (AlexNet) | LPIPS (VGG) |
|---|---|---|---|---|
| Base Conditioning | 15.0±4 | 20.5±14 | 39.8±25 | 37.0±14 |
| + Conditioning Block | 14.7±3 | 15.6±7 | 29.8±14 | 30.6±8 |
| + Stopping Criteria | 14.0±3 | 14.9±7 | 28.1±14 | 29.5±8 |
| + Object Channel | 14.1±3 | 13.3±6 | 24.9±12 | 28.6±7 |

Table 2: Ablation study of the P2P network on the tennis sequence. The results are multiplied by a factor of 1000 for readability.

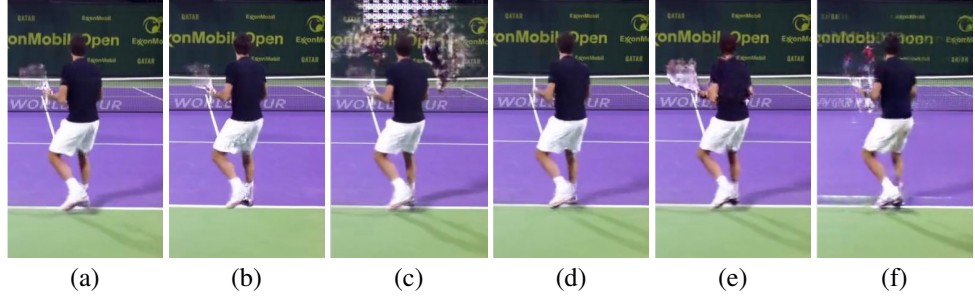

    (a)        (b)        (c)        (d)        (e)        (f)

Figure 16: P2F ablation. (a) Ours, (b) no VGG FM on the full-frame (no shadows generated), (c) no mask regularization (background artifacts), (d) 1 input pose (no racket generation due to a semantic segmentation mis-detection), (e) no discriminator FM (character/racket heavily distorted), (f) no mask, i.e. background fully-generated (excessive distortion in background / character).

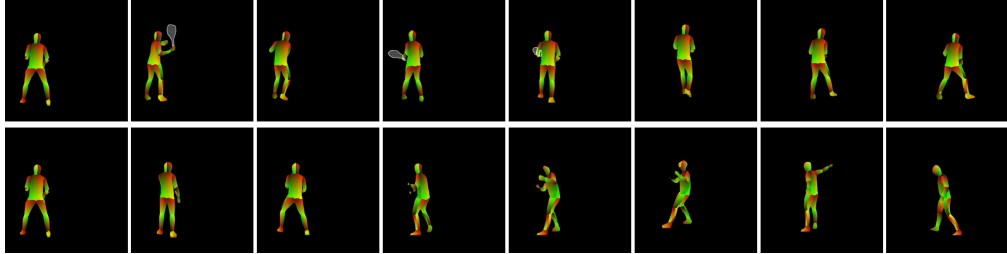

Figure 17: P2P vs. baseline method comparison. Temporal consistency of P2P generated motion (row 1) is apparent, as opposed to the baseline method (row 2), that results in temporal inconsistency.

## D ABLATION STUDY

We test the effect of several novel P2P network components, both by SSIM and LPIPS. The test is performed when predicting one frame into the future (in a "teacher forcing" mode). The results in Tab. 2 demonstrate that our conditioning block is preferable to the conventional one, and that adding the object channel is beneficial. Selecting the model based on the minimal discriminator feature-matching loss is helpful as well.

A qualitative ablation study for the P2F network is provided in Fig. 16. As can be seen, each component contributes to the naturalness of the results.

To validate the need for an autoregressive motion generation, as done by the P2P network, we implemented a baseline method that copies motion patterns from the training set, matching the displacement, and verified that such a naive approach fails to produce natural motion. A sequence of frames from the experiment video can be seen in Fig. 17.

