# OpenReview forum: "Vid2Game: Controllable Characters Extracted from Real-World Videos"
_ICLR.cc/2020/Conference — Accept (Poster)_

### Official Review · AnonReviewer2 · 2019-10-22
**Official Blind Review #2**

**Rating:** 6

**Review:**

This paper proposes a method to address the interesting task, i.e. controllable human activity synthesis, by conditioning on the previous frames and the input control signal. To synthesis the next frame, a Pose2Pose network is proposed to first transfer the input information into the next frame body structure and object. Then, a Pose2Frame network is applied to generate the final result. The results on several video sequences look nice with more natural boundaries, object, and backgrounds compared to previous methods.

Pros:
1. The proposed Pose2Pose successfully transfer the pose conditioned on the past pose and the input control signal. The proposed conditioned residual block, occlusion augmentation and stopping criteria seem to help the Pose2Pose network work well. Besides, the object is also considered in this network, which makes the method generalized well to the videos where human holds some rigid object.
2. The Pose2Frame network is similar to previous works but learns to predict the soft mask to incorporate the complex background and to produce shallow. The mask term in Eq. (7) seems to work well for the foreground (body+object) and the shallow regions.
3. The paper is easy to follow.

Cons:
1. Since the method is only evaluated on several video sequences, I am not sure how the method will perform on other different scenes. Results on more scenes will make the performance more convincing. I also wonder if the video data will be released, which could be important for the following comparisons.
2. As to the results of the Pose2Pose network, I wonder if there are some artifacts that will affect the performance of the Pose2Frame network. Then, there will be another question: how the two networks are trained? Are they trained separately or jointly? I assume the authors first train the Pose2Pose network, then use the output to train the Pose2Frame network. Otherwise, the artifacts from Pose2Pose will affect the testing performance of the Pose2Frame network.
3. The mask term seems to work well for the shallow part. I wonder how the straightforward regression term plus the smooth term will perform for the mask. Here, the straightforward regression term means directly regress the output mask to the target densepose mask. Will the proposed mask term perform better?


**Experience Assessment:**

I have published in this field for several years.

**Review Assessment: Checking Correctness Of Derivations And Theory:**

I carefully checked the derivations and theory.

**Review Assessment: Checking Correctness Of Experiments:**

I carefully checked the experiments.

**Review Assessment: Thoroughness In Paper Reading:**

I read the paper thoroughly.

---

> ### Author Response · Authors · 2019-11-05
> **Thank you for your supportive review.**
>
>
> 1. We aimed to provide a broad variety of example applications (playing tennis, walking, fencing, dancing), while mainly focusing on the most complicated (tennis) application, for a thorough analysis of our method. Unfortunately, we cannot release the dataset, since we do not own the videos, but we will share our code.
>
> 2. The Pose2Pose and Pose2Frame networks are trained separately. Specifically, the P2F network is trained on the original data, and not on the output frames of the P2P network. You are correct that some artifacts are added to the final P2F output at test time, yet they are minor due to the structural stability of the poses generated by the P2P network. Furthermore, training the P2F network with the P2P outputs is problematic, since we do not have the ground-truth for the new pose generated by the P2P network.
>
> 3. The mask loss proposed in the review is similar to our implementation, except that we make a distinction between an inner-mask control and an outer-mask control. Our mask regression losses consist of a first loss penalizing the mask from being active outside the densepose mask, and a second loss penalizing the mask from being inactive inside the densepose mask. Combining them both results in the suggested loss.

---

### Official Review · AnonReviewer1 · 2019-10-25
**Official Blind Review #1**

**Rating:** 6

**Review:**

This paper presents  a controllable model from a video of a person performing a certain
activity. It generates novel image sequences of that person, according
to user-defined control signals, typically marking the displacement of the moving
body. The generated video can have an arbitrary background, and effectively
capture both the dynamics and appearance of the person. It has two networks, Pose2Pose, and Pose2Frame. The overall pipeline makes sense; and the paper is well written.

The main problems come from the experiments, which I would ask for more things. It has two components, i.e., Pose2Pose and Pose2Frame. So how importance of each component to the whole framework? I would ask for the ablation study/additional experiments of using each component.  How about combining only Pose2Pose/ Pose2Frame  with pix2pixHD? Whether the performance can get improved?


**Experience Assessment:**

I have read many papers in this area.

**Review Assessment: Checking Correctness Of Derivations And Theory:**

I did not assess the derivations or theory.

**Review Assessment: Checking Correctness Of Experiments:**

I assessed the sensibility of the experiments.

**Review Assessment: Thoroughness In Paper Reading:**

I read the paper at least twice and used my best judgement in assessing the paper.

---

> ### Author Response · Authors · 2019-11-05
> **Thank you for your supportive review.**
>
> Pose2Pose -- An ablation study for the P2P network can be found in Table 2, with quantitative results for each contribution. We do not add a qualitative ablation study for the P2P network, since still-images (as opposed to videos) do not convey the temporal improvement in this case.
>
> Pose2Frame -- A qualitative ablation study can be found in Fig. 16. As can be seen, the results justify each component used.
>
> pix2pixHD -- the Pose2Frame network can be directly compared with the pix2pixHD network, since they both act as mapping functions between dense-pose representations to realistic images. A quantitative comparison can be found in Table 1, as well as a qualitative comparison in Fig. 14. As can be seen, the use of our different components described in the P2F ablation study (blending mask and regularization, object channel, two pose inputs, discriminator attention on character, etc.), results in much fewer artifacts, making the Pose2Frame network suitable for this application. Combining the Pose2Pose and pix2pixHD networks, would yield significant artifacts (as seen in Fig. 14), and is not suitable for this kind of application.

---

### Official Review · AnonReviewer4 · 2019-11-06
**Official Blind Review #4**

**Rating:** 6

**Review:**

The paper presents an approach to extract a character from a video and then maneuver that character in the plane, optionally with other backgrounds. The character is then redrawn into the background with a neural net, and all of this is done in real time.

All in all, this paper was well structured and extensively detailed wrt how it engineered this solution (and why). If I had a complaint, it would be that I did not learn anything scientifically from the paper. There isn't a tested hypothesis, but rather it's a feat of engineering to get this to work. Those are important as well for the field, and I suspect that this direction could be pushed a lot more. For example, it's not close to getting realistic spatial movement relative to the plane nor is the control that impressive wrt limbs. However, as a next-contribution, this work deserves to be seen more widely.

Hence, I rate it as a weak accept.

**Experience Assessment:**

I have read many papers in this area.

**Review Assessment: Checking Correctness Of Derivations And Theory:**

N/A

**Review Assessment: Checking Correctness Of Experiments:**

I assessed the sensibility of the experiments.

**Review Assessment: Thoroughness In Paper Reading:**

I made a quick assessment of this paper.

---

> ### Author Response · Authors · 2019-11-06
> **Thank you for the supportive review.**
>
> We agree with most of the comments.

---

### Decision · Program_Chairs · 2019-12-19

**Decision:**

Accept (Poster)

**Comment:**

This paper proposes to extract a character from a video, manually control the character, and render into the background in real time.  The rendered video can have arbitrary background and capture both the dynamics and appearance of the person. All three reviewers praises the visual quality of the synthesized video and the paper is well written with extensive details. Some concerns are raised. For example, despite an excellent engineering effort, there is few things the reader would scientifically learn from this paper. Additional ablation study on each component would also help the better understanding of the approach. Given the level of efforts, the quality of the results and the reviewers’ comments, the ACs recommend acceptance as a poster.